# Overlapping between Wound Healing Occurring in Tumor Growth and in Central Nervous System Neurodegenerative Diseases

**DOI:** 10.3390/brainsci13030398

**Published:** 2023-02-25

**Authors:** Domenico Ribatti, Vincenzo Benagiano, Diego Guidolin

**Affiliations:** 1Department of Translational Biomedicine and Neuroscience, University of Bari Medical School, Piazza Giulio Cesare, 70124 Bari, Italy; 2Department of Neuroscience, Section of Anatomy, University of Padova, 35122 Padova, Italy

**Keywords:** CNS neurodegenerative diseases, tumor growth, wound healing

## Abstract

Wound healing is characterized by the formation of a granulation tissue consisting of inflammatory cells, newly formed blood vessels, and fibroblasts embedded in a loose collagenous extracellular matrix. Tumors behave as wounds that fail to heal. Neuronal loss in neurodegenerative disease is associated with the synthesis and release of new components of the extracellular matrix by activated fibroblasts and astrocytes. This condition is responsible for a perpetuation of the wound healing state and constitutes a condition very similar to that which occurs during tumor progression. The aim of this article is to emphasize and compare the role of wound healing in two different pathological conditions, namely tumor growth and central nervous system neurodegenerative diseases. Both are conditions in which wounds fail to heal, as occurs in physiological conditions.

## 1. Introduction

Wound healing is a physiological process which involves cellular activation and proliferation, leading to tissue matrix remodeling. In this context, the reactive capacity of different organs is heterogeneous: skin and gut epithelia and hematopoietic cells have highly active regenerative capacities; the liver and pancreas have intermediate capacities; and the heart and central nervous system (CNS) have low or absent capacities [1].

In wound healing after injury, several different phases have been described. These include an inflammation phase involving different inflammatory cells, as well as proliferation and re-epithelization, angiogenesis, and remodeling phases, which lead to the formation of a granulation tissue. This product contains inflammatory cells, newly formed blood vessels, and fibroblasts embedded in a loose collagenous extracellular matrix [2,3] (Figure 1). Primarily, platelets form a clot, a gel that consists of fibrin, fibronectin, and platelets and this entraps plasma and blood cells. These transform into fibrin, forming an early matrix together with vitronectin and fibronectin. In a second phase, inflammatory cells which have been recruited to the anatomical site are involved, including neutrophils, macrophages, and fibroblasts.

Angiogenesis, i.e., the formation of new vessels from pre-existing ones, is required to oxygenate the new tissue and dispose of the waste products of metabolism. This mechanism is rapidly stimulated after injury, a process which ceases when wound healing is complete. Among the extracellular matrix proteins, fibronectin, a glycoprotein secreted by proliferating endothelial cells, forms the provisional extracellular matrix of wounds and promotes the migration of endothelial cells over the wound [4].

**Figure 1 brainsci-13-00398-f001:**
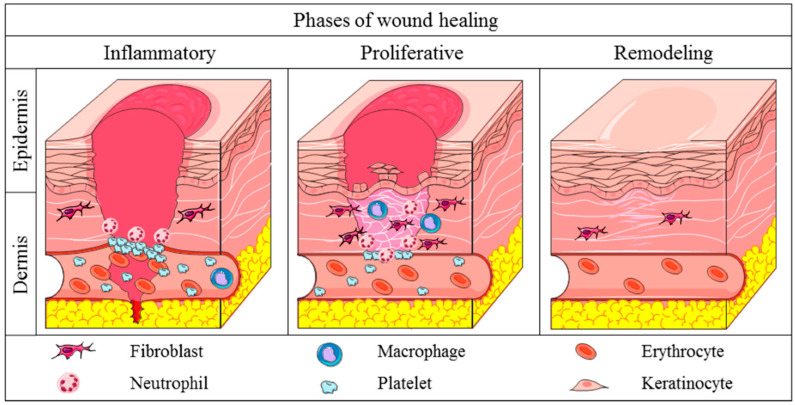
Different phases in skin wound healing (Reproduced with permission from [5]).

Neutrophils are the first inflammatory cells to move to the wound tissues. Neutrophils kill pathogens, and macrophages phagocytose neutrophils while infiltrating T cells produce interleukin (IL)-17 and IL-22 and tumor necrosis factor alpha (TNFα) to amplify the host defense response. IL-17 contributes to further recruiting neutrophils, establishing a positive loop [2]. Then, macrophages switch to an anti-inflammatory phenotype, and neutrophil T-regulator (Tregs) cells attenuate the inflammation response, facilitating resolution and tissue remodeling. In the last, remodeling phase, the wound contracts, leading to collagen IV replacing collagen I, while nerves regenerate and new vessels prune.

The aim of this article is to emphasize and compare the role of wound healing in two different pathological conditions, namely tumor growth and CNS neurodegenerative diseases, both conditions in which wounds fail to heal, as occurs in physiological conditions.

## 2. Tumors: Wounds That Do Not Heal

Tumor growth is characterized by an angiogenic response in the tumor stroma, allowing the growth of the primary tumor and favoring metastatic spread. Studying the dynamic aspects of this process the pathologist Harold F. Dvorak demonstrated similarities between tumor stroma generation and wound healing [6]. Dvorak noted that wounds, like tumors, secrete the most important angiogenic growth factor, namely vascular endothelial growth factor (VEGF), responsible for an increased vascular permeability, favoring the leak of plasma fibrinogen. This, in turn, provides a matrix to which newly formed vessels can migrate. Differently from wounds that turn off VEGF production after healing, tumors continued to make large amounts of VEGF, allowing angiogenesis to continue. Thus, tumors behave as wounds that fail to heal [6].

In skin wounds, within 24 h of wounding, VEGF mRNA expression increases in epidermal keratinocytes at the wound edge [7], reaches a peak after 2–3 days, and persists for about 1 week, as is necessary for granulation tissue to form and migrating keratinocytes to cover the wound. Then, VEGF expression is downregulated, and vascular permeability returns to normal. Keratinocyte deletion of the VEGF gene in mice resulted in delayed wound healing owing to impaired angiogenesis [8]. Moreover, the absence of VEGF signaling increases collagenase IV activity, leading to the restoration of normal basement membrane formation [9]. In tumors, this tissue has an abnormal thickness [10].

Fibrin and collagen I constitute the provisional extracellular matrix during wound healing in vivo [11,12] and, in fibrinogen-deficient mice, wound healing defect is recognizable [13]. The wound fibrin clot formed following the release of fibrinogen from vessels, together with the limited or impaired lymphatic function, retard the clearance of edema fluid and finally result in an increase in interstitial tissue pressure [14].

## 3. Wound Healing in the CNS Neurodegenerative Diseases

CNS fibrotic scar formation is a response to injury and other damage and contributes to a large degree to the pathology of CNS disorders [15,16]. For example, the fibrotic nature of multiple sclerosis (MS) is known in humans, and many fibrotic extracellular matrix molecules can be detected within the lesions [17]. Similarly, fibrotic scar formation has been observed after experimental autoimmune encephalomyelitis (EAE) induction in mice [16]. In EAE, perivascular fibroblasts are activated in the spinal cord and infiltrate the parenchyma via the peak of behavioral deficits, where they are closely associated with areas of demyelination, myeloid cell accumulation, and extracellular matrix deposition [16]. Platelet-derived growth factor receptor beta (PDGFRβ)-positive fibroblast-like cells, originating from perivascular and meningeal sources, have been found in EAE lesions and human active MS lesions [18].

### 3.1. Early Phases

Following injury, the initial phases of healing involve a response starting with an inflammatory reaction, macrophage infiltration, and secretion factors, triggering extracellular matrix deposition by fibroblasts.

Following acute injury of the CNS, the initial phases of healing are very similar [19]. The cellular debris generated by a lesion, indeed, recruits inflammatory cells and fibroblasts, leading to the formation of a so-called mesenchymal (or fibrotic) scar representing the central core of the lesion [20,21] and mainly consisting of collagen I and IV [22], as well as fibronectin and laminins [23]. The number of fibroblasts correlates with macrophage/microglia infiltration to the lesioned region [24].

In the adult healthy CNS, fibroblasts are localized in the meninges, perivascular space, and choroid plexus [25]. When an injury occurs, they are found to contribute to scar formation. In addition, fibroblast-like cells are also involved [21]. They include circulating fibrocytes migrating into the injured tissue to differentiate into fibroblasts [26], endothelial cells undergoing endothelial mesenchymal transition (EMT) [27], and a subpopulation of pericytes [28]. Fibroblast activation to secrete extracellular matrix proteins into the CNS is still not well understood (see 20 for a detailed discussion). However, it certainly involves the action of transforming growth factor beta (TGF-β) and PDGF, which are mainly secreted by macrophages/microglia. A specific feature of the initial healing process in the CNS is represented by a glial scar, surrounding the mesenchymal scar [20,21,29].

Concerning chronic injuries of the CNS, including those associated with neurodegenerative diseases, the main difference with acute injuries is the gradual accumulation of the damage and the production of an interspersed set of small individual lesions, exhibiting the above-mentioned general features in terms of cell response and extracellular matrix deposits [15,21]. Recent reviews specifically addressed the role played by healing mechanisms in the context of neurodegenerative diseases [19,21].

### 3.2. CNS Tissue Remodeling and Regeneration

During normal wound healing in various tissues, unwanted cells undergo apoptosis, and the extracellular matrix is reorganized to allow for tissue regeneration [30]. In this context, several animal and human studies (see 31 for a discussion of the topic) have indicated that central neurons are able of a regenerative response following injury. This process involves an initial degenerative event with axonal swelling and disconnection, followed by regenerative changes based on sprouting from the damaged axonal stump [31]. Between 12 and 24 h postinjury, indeed, delayed (or secondary) axotomy occurs [32]. The distal, isolated portion of the axon undergoes Wallerian degeneration, while the proximal segment retains the ability to regenerate [33]. Multiple factors are involved in this process, but the key role is played by the neurofilament protein subunits, with the misalignment of the filaments, the disruption of axoplasmic flow, and swelling [34]. Ultrastructural studies [35] indicated that the neurofilament response to injury depends on the severity of the event. Changes occurring in the axon trigger the regulation of cellular proteins in the cell body (as, for instance, actin and tubulin) towards successful axonal regeneration and synaptogenesis [36]. As a result, neurons can form axonal sprouts following an injury and a specific feature of these sprouts is the presence of GAP-43, a protein expressed at a high level during axonal outgrowth, being a major component of growth cones and presynaptic terminals [37].

However, the persistence of reactive macrophages/microglia and fibroblasts for a quite long time at the core of the lesion, with a disproportionate deposition of extracellular matrix proteins to form the mesenchymal scar, may lead to an obstacle to axonal growth in different CNS injuries and diseases [30]. Extracellular matrix proteins do not have an inhibitory effect on axons [38,39]. However, several factors inhibiting axonal growth have been reported in the scar’s matrix. They include phosphacan, neural/glial antigen 2 (NG2), tenascin-C [40], and semaphorin III [41]. In this context, hypertrophic astrocytes and activated microglia near the lesion site have been shown to be detrimental to neuron and oligodendrocyte regeneration [42].

On the other side, however, in vivo obtained evidence also exists, indicating that the glial scar, formed by astrocytes around the mesenchymal one, may help axonal growth by creating permissive bridges (called “glial bridges”) which allow growing axons to cross the fibrotic scar in response to different growth factors [43,44].

Thus, both the promotion of tissue protection and the inhibition of repair take place in the healing process of the CNS [20,21] and the response to tissue injury in the CNS could be considered as incomplete wound healing.

## 4. Concluding Remarks

A crucial event in tumor growth progression is the nonsolution of the wound healing-like process, which leads and contributes to the perpetuation of the disease and to the establishment of an altered tumor microenvironment [6,7].

The CNS displays both poor wound healing and low regenerative potential for new neurons, with the wound healing process leaving a persistent scar that impedes tissue function. The CNS scar comprises two compartmentalized cellular structures: the fibrotic scar, which is made up of fibroblasts originating from a perivascular source and infiltrating monocytes/macrophages; and the glial scar, which is composed of resident glial cell types that encase the fibrotic scar [45,46]. In acute brain and spinal cord traumas, a well-defined fibrotic scar is generated in which extracellular matrix components, myofibroblasts, and astrocytes are involved (Figure 2). Meanwhile, in neurodegenerative diseases, neuronal loss favors the progressive replacement of damaged tissue with extracellular matrix components, produced by activated fibroblasts and astrocytes, leading to a further response mediated by microglia and immune cells [47]. Overall, the tissue reaction occurring in neurodegenerative disease to compensate for neuronal loss is responsible for a perpetuation of the wound healing state, and this condition is very similar to that which occurs during tumor progression. In both tumor growth and neurodegenerative diseases, this is a fundamental aspect of explaining the perpetuation of this disease and the difficulty in developing satisfactory therapeutic treatment.

## Figures and Tables

**Figure 2 brainsci-13-00398-f002:**
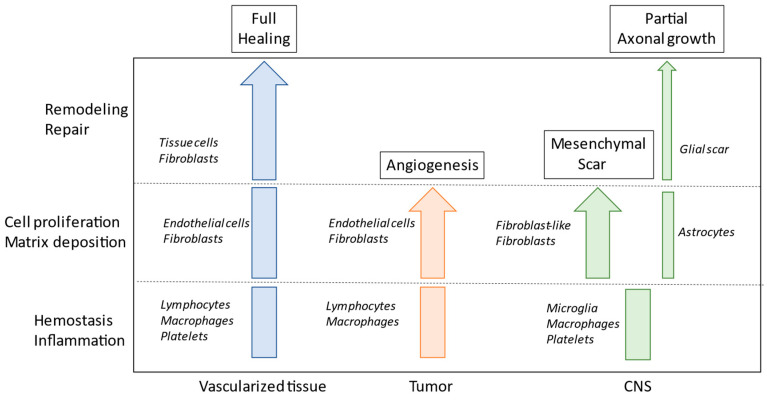
Schematic view of the process of healing in vascularized tissue, in tumors, and in central nervous system (CNS) injuries. Different cell populations are involved in the different steps.

## Data Availability

No new data were created.

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
