# Peer review of "Overlapping between Wound Healing Occurring in Tumor Growth and in Central Nervous System Neurodegenerative Diseases"

_brainsci, 2023, doi:10.3390/brainsci13030398_

Round 1
Reviewer 1 Report
The authors generalized the current research findings regarding to the relationship between wound healing and the development of cancer, and CNS injuries which may precondition the development of CNS tumor.
Overall, the manuscript is of average quality, and not well organized. The authors should focus on one major topic to expand their opinions. Either would healing and the development of tumor or would healing and the development of CNS tumor.
If the authors would like to say that the wound healing could induce cancer. And CNS injury would induce CNS wound healing and therefore may cause CNS tumor, more transitional information and paragraphs should be included.
Other minor concerns are,
1. For Figure 1, please provide the appropriate copyright information regarding to the reproduction.
2. Figure 2 is unnecessary.
Author Response
In reply to Reviewer 1
The authors generalized the current research findings regarding to the relationship between wound healing and the development of cancer, and CNS injuries which may precondition the development of CNS tumor. Overall, the manuscript is of average quality, and not well organized. The authors should focus on one major topic to expand their opinions. Either would healing and the development of tumor or would healing and the development of CNS tumor. If the authors would like to say that the wound healing could induce cancer. And CNS injury would induce CNS wound healing and therefore may cause CNS tumor, more transitional information and paragraphs should be included.
The article is not focused to establish a relationship between wound healing in tumors and the CNS injuries as precondition in the development of the CNS tumors. Our objective has been to establish a relationship between tumor growth and CNS neurodegenerative diseases, both conditions in which wounds fail to heal, as occurs in physiological conditions. In the revised version of the article, we have emphasized this aspect in the new title of the article (“Overlapping between wound healing occurring in tumor growth and in central nervous system neurodegenerative diseases”); in the Introduction section as follows: “The aim of this article is to emphasize and compare the role of wound healing in two different pathological conditions, namely tumor growth and CNS neurodegenerative diseases, both conditions in which wounds fail to heal, as occurs in physiological conditions.”, and in the Concluding remarks sections, as follows: “Overall, the tissue reaction occurring in neurodegenerative disease to compensate neuronal loss, is responsible of a perpetuation of the wound healing state, and this condition is very similar to that which occurs during tumor progression. In both tumor growth and neurodegenerative diseases, this is a fundamental aspect to explain the perpetuation of this disease and the difficulty to obtain a satisfactory therapeutic treatment.”
Other minor concerns are,
For Figure 1, please provide the appropriate copyright information regarding to the reproduction.
Done.
Figure 2 is unnecessary.
The figure 2 has been deleted.
Reviewer 2 Report
Dear Author's
Thank you for the opportunity to familiarize yourself with the content of the manuscript. I suggest minor additions, namely:
1. I do not quite understand the article title, in my opinion it needs linguistic correction
2. I suggest supplementing the text with new information regarding the presented issue
3. the entire text requires linguistic correction
4. some items of references are over 15 years old - history
5. in the text of the manuscript, some paragraphs lack references
Author Response
In reply to Reviewer 2
I suggest minor additions, namely:
I do not quite understand the article title, in my opinion it needs linguistic correction
We have changed the title of the article as follows: “Overlapping between wound healing occurring in tumor growth and in central nervous system neurodegenerative diseases”.
I suggest supplementing the text with new information regarding the presented issue
We have improved the Introduction section as follows: “The aim of this article is to emphasize and compare the role of wound healing in two different pathological conditions, namely tumor growth and CNS neurodegenerative diseases, both conditions in which wounds fail to heal, as occurs in physiological conditions” and the Concluding remarks section as follows:” Overall, the tissue reaction occurring in neurodegenerative disease to compensate neuronal loss, is responsible of a perpetuation of the wound healing state, and this condition is very similar to that which occurs during tumor progression. In both tumor growth and neurodegenerative diseases, this is a fundamental aspect to explain the perpetuation of this disease and the difficulty to obtain a satisfactory therapeutic treatment.”
The entire text requires linguistic correction
The text has been revised by a native English speaker.
Some items of references are over 15 years old – history
The article has also an historical valence to underline the scientific background of our “opinion”.
In the text of the manuscript, some paragraphs lack references
We have added new references.
Round 2
Reviewer 1 Report
Thanks for the prompt response. The article is now in good shape for publication.